# Development of a Performance Analysis Model for Free-Piston Stirling Power Convertor in Space Nuclear Reactor Power Systems

**Huaqi Li** [1,2], **Xiaoyan Tian** [2], **Li Ge** [1,*], **Xiaoya Kang** [2], **Lei Zhu** [2], **Sen Chen** [2], **Lixin Chen** [2], **Xinbiao Jiang** [2] **and Jianqiang Shan** [1]

1    School of Nuclear Science and Technology, Xi'an Jiaotong University, 28 Xianning West Road, Xi'an 710049, China; li.putin@stu.xjtu.edu.cn (H.L.); jqshan@mail.xjtu.edu.cn (J.S.)
2    Northwest Institute of Nuclear Technology, 28 Pingyu Road, Xi'an 710024, China; tianxiaoyan@nint.ac.cn (X.T.); kangxiaoya@nint.ac.cn (X.K.); zhulei@nint.ac.cn (L.Z.); chensen@nint.ac.cn (S.C.); chenlixin@nint.ac.cn (L.C.); jiangxinbiao@nint.ac.cn (X.J.)
*    Correspondence: gelili@mail.xjtu.edu.cn

**Abstract:** Space nuclear reactor power system (SNRPS) is a priority technical solution to meet the future space power requirement of high-power, low-mass, and long-life. The thermoelectric conversion subsystem is the key component of SNRPS, which greatly affects the performance, quality, and volume of SNRPS. Among all kinds of proposed thermoelectric conversion technologies, the free-piston Stirling power converter (FPSPC) has become a preferred conversion technology for small-scale advanced SNPRS due to its moderate waste heat emission temperature and high conversion efficiency, mainly composed of a linear alternator and free-piston Stirling engine (FPSE). For studying the performance of FPSPC, a quasi-steady flow thermodynamic cycle analysis model considering parasitic heat losses has been developed for FPSE. And then the performance analysis model for FPSPC has been established by coupling the thermodynamic cycle analysis model with the mechanical motion model of the piston and volt-ampere characteristic model of the linear alternator. Furthermore, the analysis model was compared and validated by the GPU-3 Stirling engine's experimental data. The performance parameters of Component Test Power Converter (CTPT) FPSPC designed by NASA for SNRPS were also analyzed. The results show that the amplitudes position of CTPC displacer and piston are 15.1 mm and 11.2 mm, respectively. The corresponding average electric power output of CTPC is 17.316 kW. The input thermal power to the CTPT heater is 66.1 kW, leading to the converter efficiency of 26.2%. The average current and voltage of the CTPC alternator are 86.38 A and 193.15 V, respectively. Among all kinds of parasitic energy losses, the regenerator heat loss accounts for the largest proportion, with an average of about 12.7 kW. The effects of cooler and heater temperature on the performance of CTPC FPSPC were also studied.

**Keywords:** free-piston Stirling power convertor; quasi-steady flow model; parasitic losses; thermal-mechanical-electrical-magnetic coupled model; performance analysis

## 1. Introduction

In the future, space exploration missions, including deep space exploration, space station, and planetary surface base will need an electric power supply of 10–1000 kW [1]. Existing conventional energy such as solar energy and chemical fuel cells cannot fully meet the development and innovation requirement of space missions in terms of long life, environmental adaptability, weight, and power density. Generally, the thermal photovoltaic cells and chemical cells are applied for low-power space missions in low Earth orbit instead of high-power or far-earth missions. Besides, solar panel cells are greatly affected by the sunlight environment, and the area of the solar panel array will increase to an unacceptable extent with the increase of power demand. For example, a 500 m$^2$ solar array would be required to produce a power of 200 kWe, which is the approximate power level of NASA's

Prometheus mission [2]. The huge array areas required for high power (>300 kWe) will cause obstacles in deployment in orbit, stowing for launch, structural integrity, and so on, which cannot be solved with current technology. Therefore, with the deep expansion of space missions, space power has become the key technology for the development of space technology [3]. To enhance the capabilities of space exploration with advanced space power technology of high-power, low-mass, and long-life, the researcher has proposed nuclear power as a source of continuous power for space missions, because the Space Nuclear Reactor Power Systems (SNRPS) have advantages of long-life, high energy density and stable working performance [4]. Specifically, the mass distribution of the SNRPS is about 20% of the reactor core subsystem, 20% of the shield subsystem, 20% of the thermoelectric conversion device subsystem, 25% of the radiant radiator, and 15% of other auxiliary subsystems [5]. The most feasible solution to reduce SNRPS's weight is improving the efficiency of the thermoelectric conversion device and reducing the area of the radiator.

Thermal-electric conversion devices for SNRPS are mainly classified into static thermal-electric conversion devices (including thermocouples (TE) and thermionic (TI)) and dynamic thermal-electric conversion devices (including Stirling cycle thermal-electric conversion and Brayton cycle thermal-electric conversion) [6]. Free-piston Stirling power convertor (FPSPC) is one of dynamic Stirling cycle thermal-electric conversion, which has the advantages of high efficiency, moderate waste heat emission temperature, minimum radiator area, minimum system mass of SNRPS, and optimal comprehensive performance in the range of 40–100 kW for SNRPs [7]. For example, NASA and the Department of Energy developed the FPS reactor concept [8] using FPSPC for its clear performance advantages under the heat source temperature of 900 K and the output power requirement of 40 kWe. FPSPC is mainly composed of a linear alternator, power control system, free-piston Stirling engine (FPSE), and power regulation and distribution system. The gas in the FPSE heater is heated by a high-temperature heat source. When the temperature reaches the starting temperature, the piston in the engine starts to vibrate, converting the thermal energy from the outside heat source into mechanical energy of the reciprocating resonant of the piston, and then into electrical energy through the linear alternator, finally realizing the conversion of thermal energy, mechanical energy, and electrical energy. The main structures of FPSPC include FPSE and linear generator, as shown in Figure 1 [9].

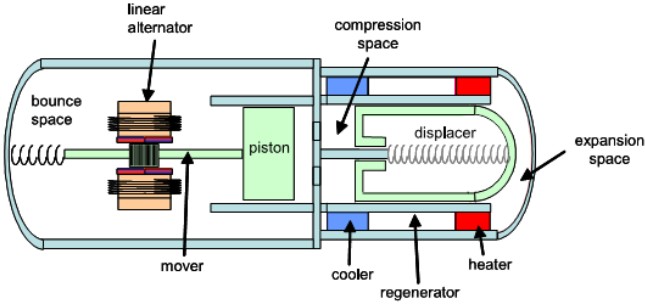

**Figure 1.** Cross-section schematic of an FPSPC.

The performance of thermal-electric conversion devices directly influences the quality, volume, and performance of SNRPS, and has an important influence on the safety characteristics of the thermal-electric conversion devices. Therefore, it is necessary to analyze the performance and transient characteristics of the FPSPC for obtaining the safety and advanced design of SNRPS. The performance analysis of FPSPC involves many subjects such as thermodynamics cycle, piston, and displacer dynamics and alternator electromagnetism, and various parasitic energy losses during the working process which are the main factors affecting the energy transfer and thermoelectric conversion efficiency of FPSPC. To come up with an efficient and feasible design of FPSPC for SNRPS, it is necessary to develop a performance analysis model integrating FPSE and linear alternator and considering the influence of major energy loss on its performance. At present, relevant researches mainly

focus on the thermodynamic analysis methods [10] for FPSE and Stirling dynamic models for use in real-time simulations of Stirling convertors [11]. However, there are few analysis models for the overall performance of FPSPC which considered the heat losses in thermodynamics analysis and linear generator volt-ampere characteristics. Therefore, based on the Urieli FPSE second-order analysis method [12], a PFSPC Stirling performance analysis model coupling thermodynamics, piston dynamics, and linear generator volt-ampere characteristic was established and corresponding analysis code FPSC_NINT (Northwest Institute of Nuclear Technology) was also developed in this paper. The model and code were validated with the GPU-3 engine's experimental data. The performance of CTPC FPSPC developed by NASA was also analyzed. It is demonstrated that developed code FPSC_NINT based on coupling performance analysis models could effectively predict the performance parameters of FPSPC.

## 2. Performance Analysis Model for FPSPC

The FPSPC involves thermal, mechanical, fluid, magnetic, and electrical domains. So far, there have been plenty of individual physical analysis models of FPSPC but few coupling models. Thus, it is necessary to develop an integrated model which combines all those individual models and is capable of the overall performance study of FPSPC.

### 2.1. The Losses and Quasi Steady Flow Model (LQSFM) for FPSE

To analyze the working process and predict the performance of FPSE, an improved Losses and Quasi Steady Flow Model (LQSFM) for FPSE internal gas circuit has been established in this paper by considering the heat loss due to different mechanisms based on the quasi-steady flow analysis thermodynamic model (QSFM) modified by Urieli.

2.1.1. Parasitic Heat Losses Model

In the LQSFM model, heat losses mainly include dissipation caused by pressure drops and internal heat conduction through heat exchangers. In addition, the gas hysteresis power loss in expansion and compression cells, and heat loss caused by the shuttle effect in the displacer have also been considered. However, seal leakage power loss and mechanical friction heat loss have been ignored.

The pressure drops occur as the working gas flows through the heat exchangers. Such pressure drops result in the reduction of the work output. The energy loss caused by the heat exchanger pressure drops ($\dot{Q}_{loss,dp}$) can be calculated as:

$$\dot{Q}_{loss,dp} = \frac{\Delta p \dot{m}}{\rho} \tag{1}$$

where $\dot{m}$ is the mass flow rate of gas (kg/s). $\rho$ is gas density (kg/m$^3$). $\Delta p$, the pressure drops in heat exchangers, are given by

$$\Delta p = -\frac{2 f_r \mu U V}{A_{free} d_h^2} \tag{2}$$

where $f_r$ is Reynolds friction coefficient; $\mu$ is gas dynamic viscosity (Pa·s). $U$ is the gas velocity (m/s). $V$ is the corresponding volume (m$^3$). $A_{free}$ is the free flow area (m$^2$). $d_h$ is the hydraulic diameter (m).

The internal conduction heat losses caused by heat conduction between the hot part of the heat exchanger and the cold part are calculated by 1-D heat conduction equation:

$$\dot{Q}_{loss,cd} = A_{eff} k \Delta T / \Delta x \tag{3}$$

where $A_{eff}$ is the area (m$^2$); $k$ is the thermal conductivity of the material (W/m·K). $\Delta T$ is the length (m). $\Delta x$ is the temperature difference (K).

The effectiveness of the regenerator is $\varepsilon = NTU/(NTU+1)$. The regenerator heat loss is defined as:

$$\dot{Q}_{loss,r} = (1-\varepsilon)\left(\dot{Q}_{r1} + \dot{Q}_{r2}\right) \tag{4}$$

where, $\dot{Q}_{loss,r}$ is the heat loss in regenerator (W). $\dot{Q}_{r1}$ and $\dot{Q}_{r2}$ is the heat transferred to the regenerator control cell (W). $NTU$ is the number of transfer units, $NTU = hA_{wg}/C_p\dot{m}$, $h$ is the overall heat transfer coefficient (W/(m²·K)). $A_{wg}$ is the regenerator internal wetted area (m²). $C_p$ is the specific heat capacity at constant pressure (J/(kg·K)).

The displacer brings a quantity of heat from the hot part to the cold part during its reciprocating motion. This energy loss is called shuttle loss, which is given by [12]

$$\dot{Q}_{loss,shtl} = \frac{0.4Z_D^2 k_D D_D}{JL_D}(T_e - T_c) \tag{5}$$

where $Z_D$, $D_D$, and $L_D$ are the stroke, diameter, and length of the displacer, respectively (m). $k_D$ is the gas thermal conductivity (W/(m·K)). $J$ is the gap length between the cylinder and the displacer (m).

The gas spring hysteresis losses of the compression and expansion cell due to the non-ideal gas pressure/volume relationship can be calculated as [13]

$$\begin{aligned}
\dot{Q}_{loss,gshc} &\cong \sqrt{\frac{1}{32}\omega\gamma^3(\gamma-1)T_{wc}\overline{p}_c k_{wc}}\left(\frac{\Delta V_c}{\overline{V}_c}\right)^2 A_{wc} \\
\dot{Q}_{loss,gshe} &\cong \sqrt{\frac{1}{32}\omega\gamma^3(\gamma-1)T_{we}\overline{p}_e k_{we}}\left(\frac{\Delta V_e}{\overline{V}_e}\right)^2 A_{we}
\end{aligned} \tag{6}$$

where $\omega$ is the operating frequency. $\gamma$ is the insulation factor. $T_{wc}$ and $T_{we}$ are the compression and expansion cell wall temperature, respectively. $\overline{p}_c$ and $\overline{p}_c$ is the expansion and compression cell average pressure, respectively. $k_{wc}$ and $k_{we}$ is the gas thermal conductivity coefficient. $\triangle V_c$ and $\triangle V_e$ are the volume amplitude in the compression space and the expansion cell, respectively. $\overline{V}_c$ and $\overline{V}_e$ is the gap spring cavity average volume. $A_{wc}$ and $A_{we}$ is the wetted area in the compression space and the expansion space, respectively.

2.1.2. The Losses and Quasi Steady Flow Model (LQSFM) of Stirling Cycle

In this work, the LQSFM model for the FPSE gas circuit has been established by considering heat losses based on the QSFM model developed by Urieli. As shown in Figure 2, the LQSFM was modified by dividing the gas circuit in FPSE into several control cells, namely compression cell, cooler cell, two cells of the regenerator, heater cell, and expansion cell. The energy and mass conservation and the ideal gas state equations were applied to each of the control cells and based on several assumptions: (1) the quality of gas is constant, that is, there is no gas leakage from sealing rings of the displacer and pistons. (2) Regarding the gas in each control cell as an ideal gas, calculate the gas temperature of different components according to the ideal gas law. (3) The pressure derivative in each control cell is the same. (4) the piston cylinders walls are adiabatic. (5) The gas temperature in different cells changes linearly. (6) The heater and cooler walls are kept isothermal at the temperature of $T_{wh}$ and $T_{wk}$. Furthermore, to analyze the system transient of SNRPS, the cooler and heater wall temperature will be solved by the energy conservation equation of lumped parameter.

According to the gas flow direction (Figure 2), the gas temperature at the compression and cooler interface, $T_{ck}$ is defined by:

$$T_{ck} = T_c, \text{ if } \dot{m}_{ck} > 0; \; T_{ck} = T_k, \text{ if } \dot{m}_{ck} \leq 0 \tag{7}$$

where $\dot{m}_{ck}$ is mass flow of gas at the compression and cooler interface (kg/s). $T_c$ and $T_k$ are the compression and cooler gas temperature, respectively (K).

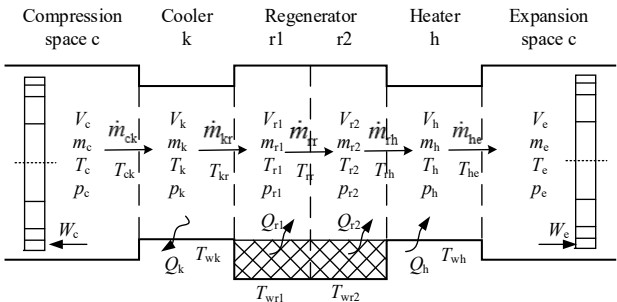

**Figure 2.** Schematic model of the engine and various temperature distributions.

Similarly, For the cooler and regenerator interface gas working fluid mass flow, the gas temperature, $T_{rk}$ is:

$$T_{kr} = T_k, \text{ if } \dot{m}_{kr} > 0; \; T_{kr} = T_{rk}, \text{ if } \dot{m}_{kr} \leq 0 \tag{8}$$

where $\dot{m}_{kr}$ is gas working fluid mass flow at the cooler and adjacent regenerator cell interface (kg/s). $T_{rk}$ is the gas at the cooler and regenerator interface (K), which can be determined by a linear extrapolation through temperature $T_{r1}$ and $T_{r2}$, $T_{rk} = (3T_{r1} - T_{r2})/2$.

Here, the regenerator is divided into two control cells $r_1$ and $r_2$ and the mixed gas mean temperature of each control cell is $T_{r1}$ and $T_{r2}$, respectively. The temperature of the gas mass flow at the boundaries of the two control cells is $T_{rr}$, which is calculated as $T_{rr} = (T_{r1} + T_{r2})/2$. the temperature of gas mass flow at the regenerator and heater interface, $T_{\rm rh}$ is:

$$T_{rh} = T_{hr}, \text{ if } \dot{m}_{rh} > 0; \; T_{rh} = T_h, \text{ if } \dot{m}_{he} \leq 0 \tag{9}$$

where $\dot{m}_{rh}$ is the regenerator and heater interface gas working fluid mass flow (kg/s). $T_{hr}$ is the gas temperature at the boundary at the regenerator and heater cell interface (K), which can be calculated as $T_{rh} = (3T_{r2} - T_{r1})/2$.

Also, the temperature of gas working fluid mass flow at the heater and expansion interface, $T_{he}$ is:

$$T_{he} = T_h, \text{ if } \dot{m}_{he} > 0; \; T_{he} = T_e \text{ if } \dot{m}_{he} \leq 0 \tag{10}$$

where $\dot{m}_{he}$ is the gas working fluid mass flow at the heater and expansion interface (kg/s); $T_h$ and $T_e$ are the heater and expansion gas temperature, respectively (K).

Considering the heat loss of gas spring hysteresis cells, heat loss of the heat exchangers and the other losses, the energy equations of all control cells are obtained:

$$-C_p T_{ck} \dot{m}_{ck} = \frac{d}{dt} W_c - \dot{Q}_{loss,gshc} + C_V \frac{d(m_c T_c)}{dt} \tag{11}$$

$$\dot{Q}_k - \dot{Q}_{loss,dpk} + C_p T_{ck} \dot{m}_{ck} - C_p T_{kr} \dot{m}_{kr} = C_V \frac{d(m_k T_k)}{dt} \tag{12}$$

$$\dot{Q}_{r1} - \dot{Q}_{loss,dpr1} + C_p T_{kr} \dot{m}_{kr} - C_p T_{rr} \dot{m}_{rr} = C_V \frac{d(m_{r1} T_{r1})}{dt} \tag{13}$$

$$\dot{Q}_{r2} - \dot{Q}_{loss,dpr2} + C_p T_{rr} \dot{m}_{rr} - C_p T_{rh} \dot{m}_{rh} = C_V \frac{d(m_{r2} T_{r2})}{dt} \tag{14}$$

$$\dot{Q}_h - \dot{Q}_{loss,dph} + C_p T_{rh} \dot{m}_{rh} - C_p T_{he} \dot{m}_{he} = C_V \frac{d(m_h T_h)}{dt} \tag{15}$$

$$C_p T_{he} \dot{m}_{he} - \dot{Q}_{loss,shtl} = \frac{d}{dt} W_e + C_V \frac{d(m_e T_e)}{dt} - \dot{Q}_{loss,gshe} \tag{16}$$

Here, $m_c$ and $m_h$ are the gas mass in the cooler and heater cells, respectively (kg); $\dot{Q}_{loss,dpk}$, $\dot{Q}_{loss,dpr1}$, $\dot{Q}_{loss,dpr2}$, and $\dot{Q}_{loss,dph}$ are the heat loss caused by the pressure drops in the cooler cell, the first part of regenerator cell, the second part of regenerator cell and heater cell, respectively (W).

The work done by or to the gas in the expansion and compression cells are:

$$dW_c/dt = p_c dV_c/dt, dW_e/dt = p_e dV_e/dt \tag{17}$$

Since pressure in the FPSE of each control cell is variable, the compression gas pressure $p_c$ is chosen as the reference pressure in the LQSFM model. For each step of the solution, $p_c$ is determined by the differential equation and each cell pressure is then calculated by $p_c$. Therefore, the transient gas pressure in each cell can be calculated as:

$$p_k = p_c + \frac{\Delta p_k}{2} \; , \; p_{r1} = p_k + \frac{(\Delta p_k + \Delta p_{r1})}{2}$$
$$p_{r2} = p_{r1} + \frac{(\Delta p_{r1} + \Delta p_{r2})}{2} \; , \; p_h = p_{r2} + \frac{(\Delta p_{r2} + \Delta p_h)}{2} \; , \; p_e = p_h + \frac{\Delta p_h}{2} \tag{18}$$

where $\Delta p_k$ and $\Delta p_h$ is the cooler and heater cell's pressure drop, respectively, (Pa).

Substitute the state equation and the associated ideal gas correlation such as $R = C_p - C_v$ and $\gamma = C_p / C_v$ into Equations (11)~(16) and then simplify the equations as follows:

$$-C_p T_{ck} \dot{m}_{ck} = \frac{1}{R} \left( C_p p_c \frac{dV_c}{dt} + C_v V_c \frac{dp_c}{dt} \right) - \dot{Q}_{loss,gshc} \tag{19}$$

$$\dot{Q}_k - \dot{Q}_{loss,dpk} + C_p T_{ck} \dot{m}_{ck} - C_p T_{kr} \dot{m}_{kr} = \frac{C_v V_k}{R} \frac{dp_c}{dt} \tag{20}$$

$$\dot{Q}_{r1} - \dot{Q}_{loss,dpr1} + C_p T_{kr} \dot{m}_{kr} - C_p T_{rr} \dot{m}_{rr} = \frac{C_v V_{r1}}{R} \frac{dp_c}{dt} \tag{21}$$

$$\dot{Q}_{r2} - \dot{Q}_{loss,dpr2} + C_p T_{rr} \dot{m}_{rr} - C_p T_{rh} \dot{m}_{rh} = \frac{C_v V_{r2}}{R} \frac{dp_c}{dt} \tag{22}$$

$$\dot{Q}_h - \dot{Q}_{loss,dph} + C_p T_{rh} \dot{m}_{rh} - C_p T_{he} \dot{m}_{he} = \frac{C_v V_h}{R} \frac{dp_c}{dt} \tag{23}$$

$$C_p T_{he} \dot{m}_{he} - \dot{Q}_{loss,shtl} = \frac{1}{R} \left( C_p p_e \frac{dV_e}{dt} + C_v V_e \frac{dp_e}{dt} \right) - \dot{Q}_{loss,gshe} \tag{24}$$

The pressure derivative can be obtained by adding up all energy conservation equations above:

$$\frac{dp_c}{dt} = \frac{1}{C_v V_t} \left[ R \left( \dot{Q} - \dot{Q}_{loss,dp} \right) - C_p \frac{\delta W}{dt} \right] \tag{25}$$

$$\dot{Q} = \dot{Q}_k + \dot{Q}_{r1} + \dot{Q}_{r2} + \dot{Q}_h - \dot{Q}_{loss,shtl} \tag{26}$$

$$\dot{Q}_{loss,dp} = \dot{Q}_{loss,dpk} + \dot{Q}_{loss,dpr1} + \dot{Q}_{loss,dpr2} + \dot{Q}_{loss,dph} \tag{27}$$

$$\frac{dW}{dt} = P_c \frac{dV_c}{dt} + P_e \frac{dV_e}{dt} - \dot{Q}_{loss,gshc} - \dot{Q}_{loss,gshe} \tag{28}$$

The mass flow in the different engine components is given by the expanded energy conservation Equations (19)~(24) and Equations (25)~(28):

$$\dot{m}_{ck} = -\frac{1}{R T_{ck}} \left( p_c \frac{dV_c}{dt} + \frac{V_c}{\gamma} \frac{dp_c}{dt} \right) + \frac{\dot{Q}_{loss,gshc}}{C_p T_{ck}} \tag{29}$$

$$\dot{m}_{kr} = \frac{1}{C_p T_{kr}} \left( \dot{Q}_k - \dot{Q}_{loss,dpk} + C_p T_{ck} \dot{m}_{ck} - \frac{C_v V_k}{R} \frac{dp_c}{dt} \right) \tag{30}$$

$$\dot{m}_{rr} = \frac{1}{C_p T_{rr}} \left( \dot{Q}_{r1} - \dot{Q}_{loss,dpr1} + C_p T_{kr} \dot{m}_{kr} - \frac{C_v V_{r1}}{R} \frac{dp_c}{dt} \right) \tag{31}$$

$$\dot{m}_{rh} = \frac{1}{C_p T_{rh}} \left( \dot{Q}_{r2} - \dot{Q}_{loss,dpr2} + C_p T_{rr} \dot{m}_{rr} - \frac{C_v V_{r2}}{R} \frac{dp_c}{dt} \right) \tag{32}$$

$$\dot{m}_{he} = \frac{1}{C_p T_{he}} \left( \dot{Q}_h - \dot{Q}_{loss,dph} + C_p T_{rh} \dot{m}_{rh} - \frac{C_v V_h}{R} \frac{dp_c}{dt} \right) \tag{33}$$

The mass conservation equation of working gas for each control cell can be presented as:

$$\dot{m}_c = -\dot{m}_{ck} \tag{34}$$

$$\dot{m}_k = \dot{m}_{ck} - \dot{m}_{kr} \tag{35}$$

$$\dot{m}_{r1} = \dot{m}_{kr} - \dot{m}_{rr} \tag{36}$$

$$\dot{m}_{r2} = \dot{m}_{rr} - \dot{m}_{rh} \tag{37}$$

$$\dot{m}_h = \dot{m}_{rh} - \dot{m}_{he} \tag{38}$$

Additionally, the heat transfer rate of each regenerative heat exchanger can be calculated as:

$$\dot{Q} = h A_w \left( T_w - T_f \right) \tag{39}$$

where $h$ is the heat transfer coefficient determined by Colburn's correlation [14]. Considering the internal conduction heat losses in the regenerator and heat exchangers, the energy exchanged in the heater and cooler are given by

$$\dot{Q}_k = h_k A_{wk} (T_{wk} - T_k) - \dot{Q}_{loss,cdk} \tag{40}$$

$$\dot{Q}_h = h_h A_{wh} (T_{wh} - T_h) - \dot{Q}_{loss.cdh} \tag{41}$$

where $\dot{Q}_{loss,cdk}$ and $\dot{Q}_{loss.cdh}$ are the internal conduction heat losses of the heater and cooler (W), respectively. The energy exchanged in each cell of the regenerator is determined as:

$$\dot{Q}_{r1} = \varepsilon h_{r1} A_{wr1} (T_{wr1} - T_{r1}) - \dot{Q}_{loss,cdr1}/2 \tag{42}$$

$$\dot{Q}_{r2} = \varepsilon h_{r2} A_{wr2} (T_{wr2} - T_{r2}) - \dot{Q}_{loss,cdr2}/2 \tag{43}$$

Thus, the energy conservation equation for the regenerator matrix can be written as:

$$\frac{dT_{wr1}}{dt} = -\frac{\dot{Q}_{r1}}{C_{mr}} , \frac{dT_{wr2}}{dt} = -\frac{\dot{Q}_{r2}}{C_{mr}} \tag{44}$$

The changes of pressure, temperature, mass flow rate, heat transfer, heat loss, and external power of the FPSE system can be obtained by solving the above-coupled equations. Then, the change of the compression and expansion space volumes in the above equations were determined using mechanical dynamics equations of the displacer and piston.

### 2.2. Piston Mechanical Dynamics Analysis Model

The FPSPC combines the mechanical dynamics of the displacer and power piston based on the differential equations derived from the Mass-spring-damping system model as shown in Figure 3. The forces acting on the displacer and power piston include pressure cave forces, bending spring forces, damping forces, load forces, and bounce space pressures forces.

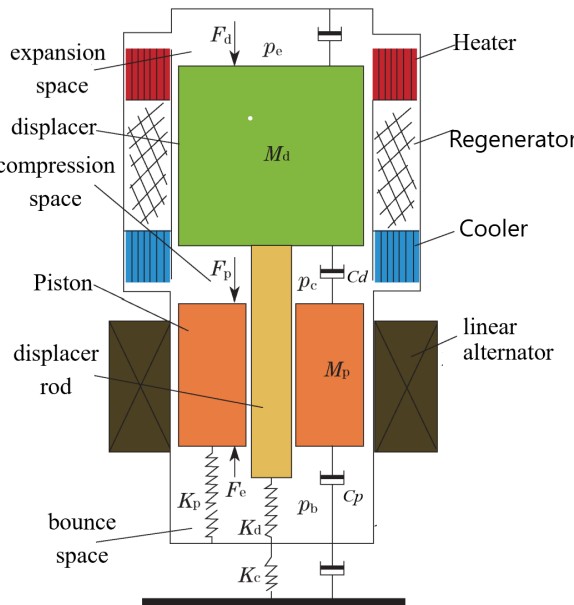

**Figure 3.** Schematic of the mass-spring-damper model of FPSPC.

According to the one-dimension translation mechanical system [15], the equations of the power piston for FPSPC can be obtained:

$$\begin{aligned}
&M_p \ddot{x}_p + c_p \dot{x}_p + K_p x_p = F_p + F_e \\
&F_p = \Delta p A_p = (p_b - p_c) A_p \\
&F_e = N \frac{d\Phi}{dx_p} \frac{1}{\eta_{mag}} I_{alt} = B L I_{alt} = K_i I_{alt}
\end{aligned} \tag{45}$$

where $M_p$ is the power piston mass (kg). $x_p$ is the power piston displacement (m). $c_p$ is the power piston damping coefficient (N·s /m). $K_p$ is the gas spring constant (N/m). $\Delta p$ is the gas differential pressure between the backpressure space and compression cell (Pa). $A_p$ is the power piston cross-sectional area (m$^2$). $B$ is the magnetic induction intensity of the linear generator ($T$). $L$ is the coil length, (m). $N$ is the number of turns of the generator winding; $\Phi$ is the magnetic flux (Wb). $\eta_{mag}$ is the generator magnetic efficiency. $I_{alt}$ is the generator current (A). $K_i$ is the alternator current electromagnetic force constant (N/A). Then the differential equation of power piston position and velocity is:

$$\begin{aligned}
&\frac{dx_p}{dt} = \dot{x}_p \\
&\frac{d\dot{x}_p}{dt} = -\frac{K_p}{M_p} x_p - \frac{c_p}{M_p} \dot{x}_p + \frac{K_i}{M_p} I_{alt} + \frac{A_p}{M_p} (p_b - p_c)
\end{aligned} \tag{46}$$

The same analysis can be performed on the FPSPC displacer-mover assembly. The movement of the displacer is mainly affected by the aerodynamic force and the spring force of the leaf spring. The movement of the power piston is then affected by the change of the aerodynamic force and airflow damping caused by the movement of the displacer. According to the one-dimension translation mechanical system, the equations of the displacer for FPSPC can be obtained:

$$\begin{aligned}
&M_d \ddot{x}_d + c_d \dot{x}_d + K_d x_d = F_d \\
&F_d = -p_e A_d + p_c (A_d - A_r) + p_b A_r = \Delta p A_r + \Delta p_R A_d
\end{aligned} \tag{47}$$

where $M_d$ is the displacer mass (kg). $x_d$ is the displacer piston displacement (m). $c_d$ is the displacer damping coefficient (N·s/m). $K_d$ is the gas spring constant (N/m). $\Delta p_R$ is the gas differential pressure between the compression and expansion cell (Pa). $A_r$ and $A_d$ are

the cross-sectional areas of the displacer and its rod (m$^2$). Then the differential equation of displacer position and velocity is:

$$\frac{dx_d}{dt} = \dot{x}_d$$
$$\frac{d\dot{x}_d}{dt} = -\frac{K_d}{M_d}x_d - \frac{c_d}{M_d}\dot{x}_d + \frac{A_r}{M_d}(p_b - p_c) + \frac{A_d}{M_d}(p_c - p_e) \tag{48}$$

Then, the compression and expansion space volumes $V_e$ and $V_c$ can be calculated as:

$$V_e = V_{eo} - A_d x_d$$
$$V_c = V_{co} - A_p x_p + (A_d - A_{rod})x_d \tag{49}$$

Also, the change of the compression and expansion cell volumes are:

$$\frac{dV_e}{dt} = -A_d \dot{x}_d$$
$$\frac{dV_c}{dt} = -A_p \dot{x}_p + (A_d - A_{rod})\dot{x}_d \tag{50}$$

The displacement and velocity of the FPSPC power piston and displacer can be obtained using the above equations. The equations of compression and expansion cell volumes need to be coupled with the LQSFM model and alternator analysis model.

### 2.3. Alternator Analysis Model

With the temperatures and pressures known, the last parameter needed for determining the piston dynamics is piston damping caused by the alternator. Hence, a simplified alternator system analysis model is established for the FPSPC alternator and load electrical circuit as shown in Figure 4. The piston-alternator configuration consists of the power piston, magnets that are attached to the power piston and wire coils that surround the piston magnets. The magnets are very close to the coils, which allowed the alternator voltage $v_{emf}$ to be calculated by a linear correlation to the piston velocity as follows [16]:

$$v_{emf} = K_e \cdot \dot{x}_p, \text{ where } K_e = N\frac{d\Phi}{dx_p} \tag{51}$$

where $K_e$ is the alternator constant (V·s/m).

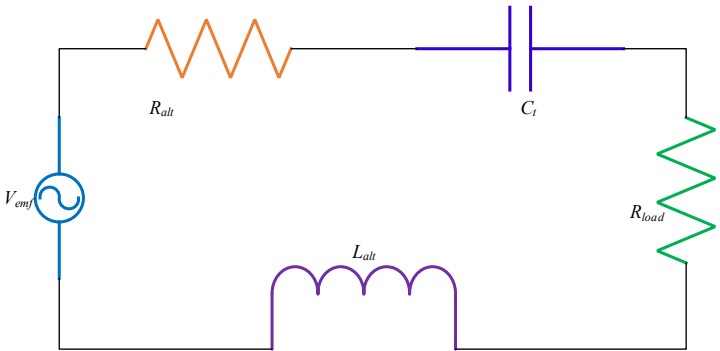

**Figure 4.** Schematic of the FPSPC alternator and load circuit linear model.

Voltage $v_{emf}$ induces a current in the coils that must be calculated by a second-order equation to describe the electrical circuit (Figure 4), which is given by:

$$v_{emf} = v_{R_{alt}} + v_{L_{alt}} + v_{C_t} + v_{R_{load}} \tag{52}$$

Substituting each voltage formula into the above equation, the formula of the alternating current is sorted out as:

$$\frac{dI_{alt}}{dt} = \frac{K_e}{L_{alt}}\dot{x}_p - \frac{R_{alt}+R_{load}}{L_{alt}}I_{alt} - \frac{1}{L_{alt}}v_{C_t}$$
$$\frac{dv_{C_t}}{dt} = \frac{1}{C_t}I_{alt}$$

(53)

where $L_{alt}$ is the generator inductance (H). $R_{alt}$ is the generator resistance ($\Omega$). $R_{load}$ is the external load resistance ($\Omega$). $C_t$ is the tuning capacitance ($\mu$F).

Using the current and voltage solved by Equations (51) and (53), the electrical power produced by FPSPC was calculated as follows:

$$\dot{W}_e = \eta_{alt}I_{alt}v_{emf}$$

(54)

where $\dot{W}_e$ is the output power of FPSPC (W). $\eta_{alt}$ is the efficiency of the alternator.

The net convertor efficiency of FPSPC is then calculated as follows.

$$\eta_{conv} = \frac{\dot{W}_e}{\dot{Q}_h}$$

(55)

where $\eta_{conv}$ is the transient convertor efficiency of FPSPC. The average convertor efficiency was determined by integrating the electrical power over time and dividing the result by the time integral of the thermal power added to the FPSPC.

## 3. Solution and Validation of the Model

### 3.1. Solution of the Model

By coupling the above LQSFM model, piston mechanical dynamics analysis model and alternator analysis model together, the independent differential equations are obtained and solved simultaneously for the variables $p_c$, $m_c$, $T_{wr}$, $x_d$, $x_p$, $I_{alt}$, $v_{emf}$, etc. Make the vector $Y$ collectively represent all unknown variables, thus $Y[p_c]$ is the compression space pressure, $Y[x_p]$ is the piston position, and so on. Given an initial condition $Y(t = 0) = Y_0$ and the corresponding differential equations $dY = F(t, Y)$, evaluate the unknown functions $Y(t)$ that satisfy both the initial conditions and the differential equations. In this problem, an implicit iterative numerical solution is applied. First, compute the derivative's value of all the variables at $t_0$. Then proceed the calculation to a new time $t_1 = t_0 + \Delta t$ by small increments of $\Delta t$ till the end of the calculation. For each time step, iteration is needed to make the calculation converge.

After that, all the unknown variables at different times will be obtained. To improve the convergence speed of the coupled model, the Schmidt model is used to determine the initial temperature of expansion and compression cells. The equations are solved numerically using the Runge-Kutta method [17]. Based on the above models and numerical solution method, the performance analysis code FPSC_NINT of space reactor FPSPC was developed in FORTRAN. The specific code calculation flow chart is shown in Figure 5.

### 3.2. Validation of the Model by GPU-3 Engine

To validate the improved LQSFM model and the FPSC_NINT code developed in this paper, the simulation results were compared with the GPU-3 Stirling engine experimental data. The GPU-3 engine consists of a tubular heater and cooler and a rhombic driving mechanism as shown in Figure 6. The main design parameters of the GPU-3 Stirling engine are listed in Table 1 [18].

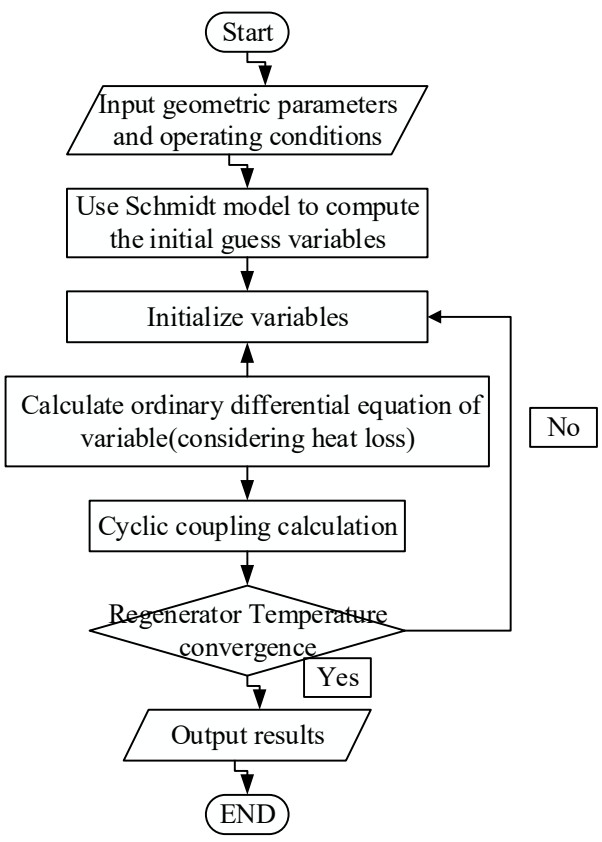

**Figure 5.** Flow chart of FPSC_NINT code calculation.

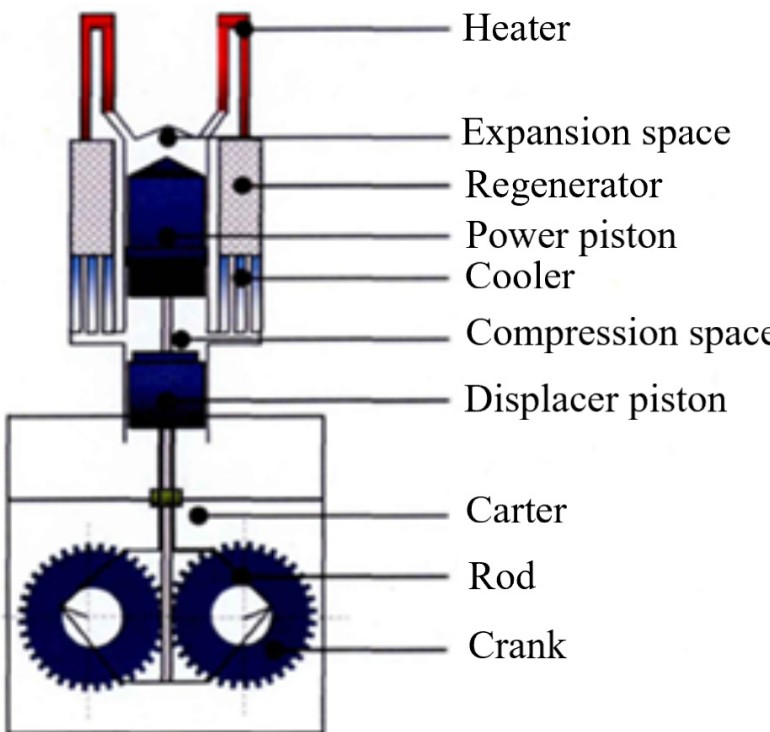

**Figure 6.** Schematic of GPU-3 engine.

Instead of ignoring the loss at the beginning, the model gradually introduces the loss and develops it gradually. The comparison between the test data and the results of each model is

listed in Table 2. The average power output of the GPU-3 engine is equal to 5.48 kW. The average heat absorbed by the heater is 13.14 kW, which leads to an engine efficiency of 40.8%. Compared with the Urieli adiabatic model, the power and the efficiency calculated by the LQSFM model which considers all losses are closer to the test date of the GPU-3 engine. We note that they are a litter different from the test date. This is probably due to different settings of equations and the models ignored the seal leakage power loss and mechanical friction between moving parts, such as appendix gap and displacer conduction losses.

**Table 1.** GPU-3 engine design parameters.

| Parameter | Value |
|---|---|
| Type | Beta |
| Heat temperature $T_h$, K | 977 |
| Heat sink temperature $T_c$, K | 288 |
| Mean effective pressure, MPa | 4.13 |
| Working gas | Helium |
| Working frequency, Hz | 41.7 |
| Regenerator diameter(inside), mm | 22.6 |
| Regenerator length, mm | 22.6 |
| Number of regenerators per cylinder | 8 |
| Regenerator wire diameter, mm | 0.04 |
| Matrix number of wires, mm | $7.9 \times 7.9$ |
| Displacer diameter, mm | 69.6 |

**Table 2.** Comparative validation of LQSFM with GPU-3 engine.

| Model | Heat (J/Cycle) | Power Output | | Efficiency (%) |
|---|---|---|---|---|
| | | W | J/Cycle | |
| Urieli Quasi-steady flow model [14] (pressure drop including $\dot{Q}_{loss,dp}$) = W1 | | 6700 | | 52.5 |
| W1 + Heat conduction loss $\dot{Q}_{loss,cd}$ = (W2) | 299.4 | 6548.3 | 157.0 | 52.4 |
| W2 + Regenerator heat transfer loss $\dot{Q}_{loss,r}$ = (W3) | 299.6 | 6546.6 | 156.9 | 52.4 |
| W3 + Shuttle heat loss $\dot{Q}_{loss,shtl}$ = (W4) | 369.1 | 6238.5 | 149.5 | 40.5 |
| W4 + Gas spring hysteresis loss $\dot{Q}_{loss,gsh}$ = LQSFM | 321.7 | 5481.7 | 131.4 | 40.8 |
| GPU-3 test data | | 3958 | | 35 |

The heat flow rate, temperature, pressure drop, and heat loss for each cell versus crank angle ($\theta = 2\pi f t$) is shown in Figure 7. The results show that parasitic heat loss must be considered to analyze the performance of the Stirling engine cycle. The internal conduction heat loss in the cooler and heater is negligible and while it is about 8.6 kW in the regenerator. This is because the axial temperature variation in the regenerator is very significant. The energy dissipation mainly occurs in the regenerator with an average power of 896 W and a maximum power of 3.4 kW, as shown in Figure 7d. The average energy dissipation in the heater and cooler are 24 W and 126 W, respectively. The average heat loss from the shuttle effect is about 3.4 kW. The heat loss due to the efficiency of the regenerator is about 1.2 kW.

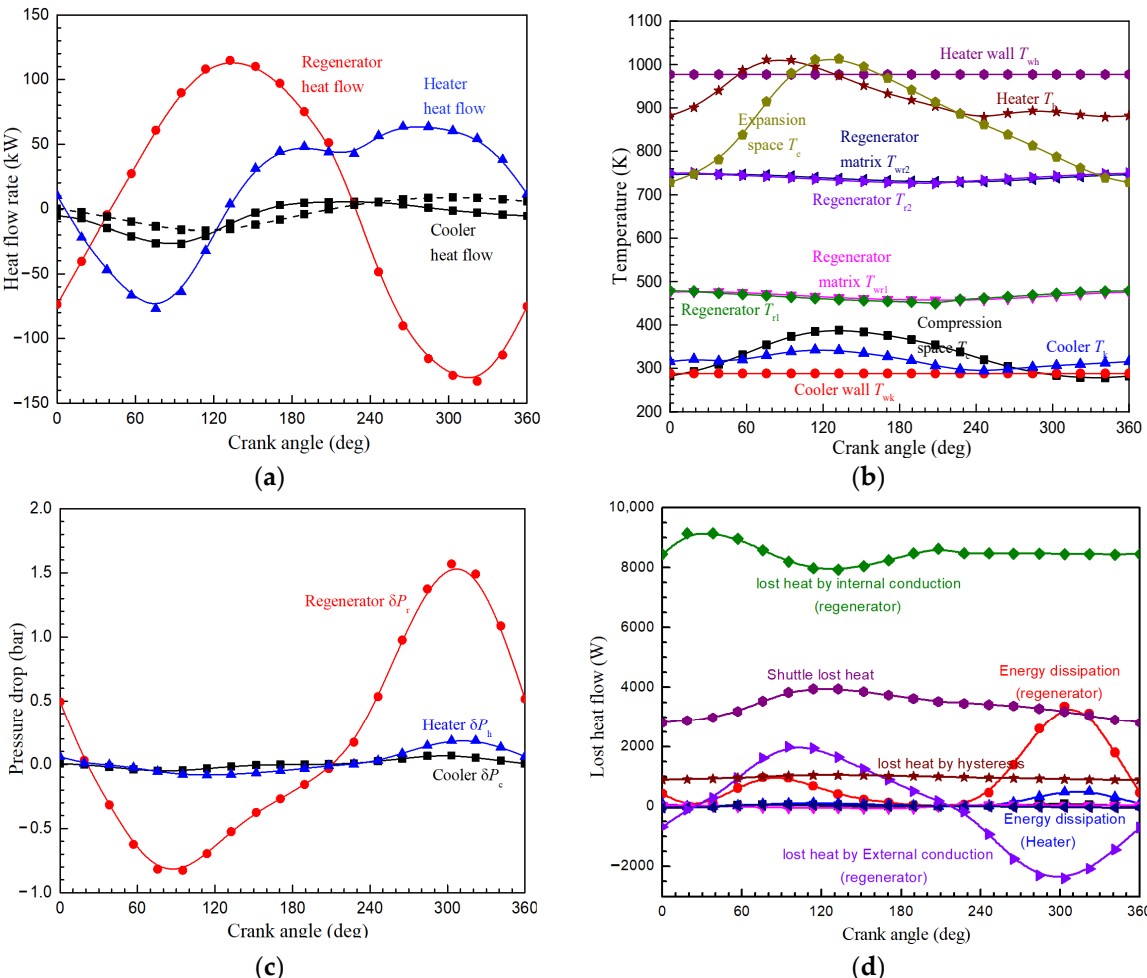

**Figure 7.** Result of the LQSFM model for the GPU-3: (**a**) the heat flow rate, (**b**) the temperature, (**c**) the pressure drop, (**d**) the heat loss.

## 4. Performance Analysis of CTPC

The CTPC is a multi-kilowatt free-piston Stirling convertor that was tested by Mechanical Technology Inc. (MTI, Latham, NY, USA) in the late 1980s and early 1990s [19]. The heat of the convertor was transferred from radiant electric heaters or heat pipes and converted into electric power of about 70-Hz AC, with the heater and cooler temperature of 800 K and 400 K, respectively. However, the CTPC represents hardware that was developed in the late 1980s, so there are few available test data. Thus, the performance parameters of CTPC under steady-state operation were analyzed using the program FPSC_NINT. For that purpose, CTPC was modeled by the coupling model FPSPC based on parameters available in a NASA document. Key model parameters are provided in Table 3.

### 4.1. Design Performance Analysis of CTPC

A comparison of CTPC design parameters, test data, and FPSC_NINT simulation results are summarized in Table 4. It is noted that it is a little different from those of CTPC design parameters and test data. This is probably due to using different parameters in the model of the alternator and the lack of operating conditions about the test. During CTPC simulation, the external load resistance is assumed to equate to alternator resistance at operating temperature, which can easily be incorporated into the model when more data become available. The external load resistance has an important influence on the voltammetry characteristics of CTPC, which may result in errors between simulation results and experimental data.

**Table 3.** The design parameters of CTPC.

| Parameters | Value | Parameters | Value |
|---|---|---|---|
| Block number | 1 | Displacer | |
| type | $\gamma$ | Material | Inconel 718 |
| Working fluid | Helium | diameter (hot) | $1.143 \times 10^{-1}$ m |
| Frequency | 70 Hz | area (hot) | $1.0261 \times 10^{-2}$ m$^2$ |
| Mean pressure | 15 MPa | area (cold) | $9.7902 \times 10^{-3}$ m$^2$ |
| Heater wall temperature | 800 K | The effective area of connecting rod | $4.708 \times 10^{-4}$ m$^2$ |
| Cooler wall temperature | 400 K | Length | $3.764 \times 10^{-2}$ m |
| Piston and displacer amplitude | 14 mm | Piston | |
| Phase angle | 67° | Material | Beryllium |
| Power out | 12.5 kW | Diameter | $1.3716 \times 10^{-1}$ m |
| Convertor efficiency | >20% | Area | $1.4776 \times 10^{-2}$ m$^2$ |
| Expansion space | | Regenerator | |
| Wall material | Inconel 718 | Porous materials | SS347 |
| Mean surface area | $7.273 \times 10^{-2}$ m$^2$ | Wall materials | Inconel 718 |
| Dead volume | $2.2982 \times 10^{-4}$ m$^2$ | Outside diameter | $2.278 \times 10^{-1}$ m |
| Average volume | $4.279 \times 10^{-4}$ m$^2$ | Inside diameter | $1.169 \times 10^{-1}$ m |
| Heater | | Wire diameter | $5.08 \times 10^{-5}$ m |
| Number of tubes | 1900 | Length | $3.76 \times 10^{-2}$ m |
| Tube inside diameter | $1.016 \times 10^{-3}$ m | Porosity | 0.728 |
| Tube length | $5.969 \times 10^{-2}$ m | Outside wall thickness | $3.175 \times 10^{-4}$ m |
| tube wall thickness | $7.5 \times 10^{-4}$ m | Inside wall thickness | $1.27 \times 10^{-3}$ m |
| Cooler | | Compression space | |
| Number of ducts | 2580 | Wall material | Beryllium |
| Duct height | $1.464 \times 10^{-3}$ m | Mean surface area | $9.442 \times 10^{-2}$ m$^2$ |
| Duct width | $5.334 \times 10^{-4}$ m | Dead volume | $1.498 \times 10^{-4}$ m$^3$ |
| Duct length | $7.493 \times 10^{-2}$ m | Average volume | $6.3529 \times 10^{-4}$ m$^3$ |

The position of displacer and piston, volumes, and gas pressure of the expansion and compression versus time are shown in Figure 8. It is found from the oscillations in Figure 8 that CTPC is operating close to the de-signed frequency of 70 Hz. As demonstrated in Figure 8a, the magnitude of the piston movement, is 11.2 mm, just a little smaller than the designed 14 mm, while that of the displacer movement is 15.1 mm, a little bigger than the designed 14 mm. The mean pressure of compression is much close to the designed 150 bar.

The heat flow rate and temperature for each cell versus crank angle are shown in Figure 9. The corresponding average power of the FPSE is 23.958 kW and the average heat energy absorbed by the heater is 66.1 kW, which leads to an engine efficiency of 36.24%. The average temperatures of heater, regenerator, and cooler are 405.2 K, 587.6 K, and 776.9 K, respectively. The pressure drop is mainly observed in the regenerator, which is the largest in all heat exchangers with a maximum of 0.5 bar and an average of 0.29 bar. In the heater and cooler, it is 0.09 bar and 0.12 bar, respectively.

**Table 4.** CTPC model simulation results versus test data.

| Parameter | Design Value | 800 K Test Data | FPSC Simulation | Error vs Test Data |
|---|---|---|---|---|
| Power out, W | 12,500 | 12,780 | 17,316 | 35% |
| Current, A | - | 48.09 | 86.38 | 79% |
| Voltage, V | - | 401.0 | 193.15 | −51.8% |
| Frequency, Hz | 70 | 67.45 | 69 | 2.3% |
| $x_d$ amplitude, mm | 14 | 14.8 | 15.1 | 2.03% |
| $x_p$ amplitude, mm | 14 | 13.44 | 11.2 | −16.67% |
| Mean pressure, bar | 150 | 150 | 149.3 | −0.47% |
| $T_h$, K | 800 | 800 | 800 | - |
| $T_e$, K | - | 776 | 756 | −2.6% |
| $T_c$, K | - | 418.5 | 405 | −3.22% |
| $T_k$, K | 400 | 400 | 400 | - |

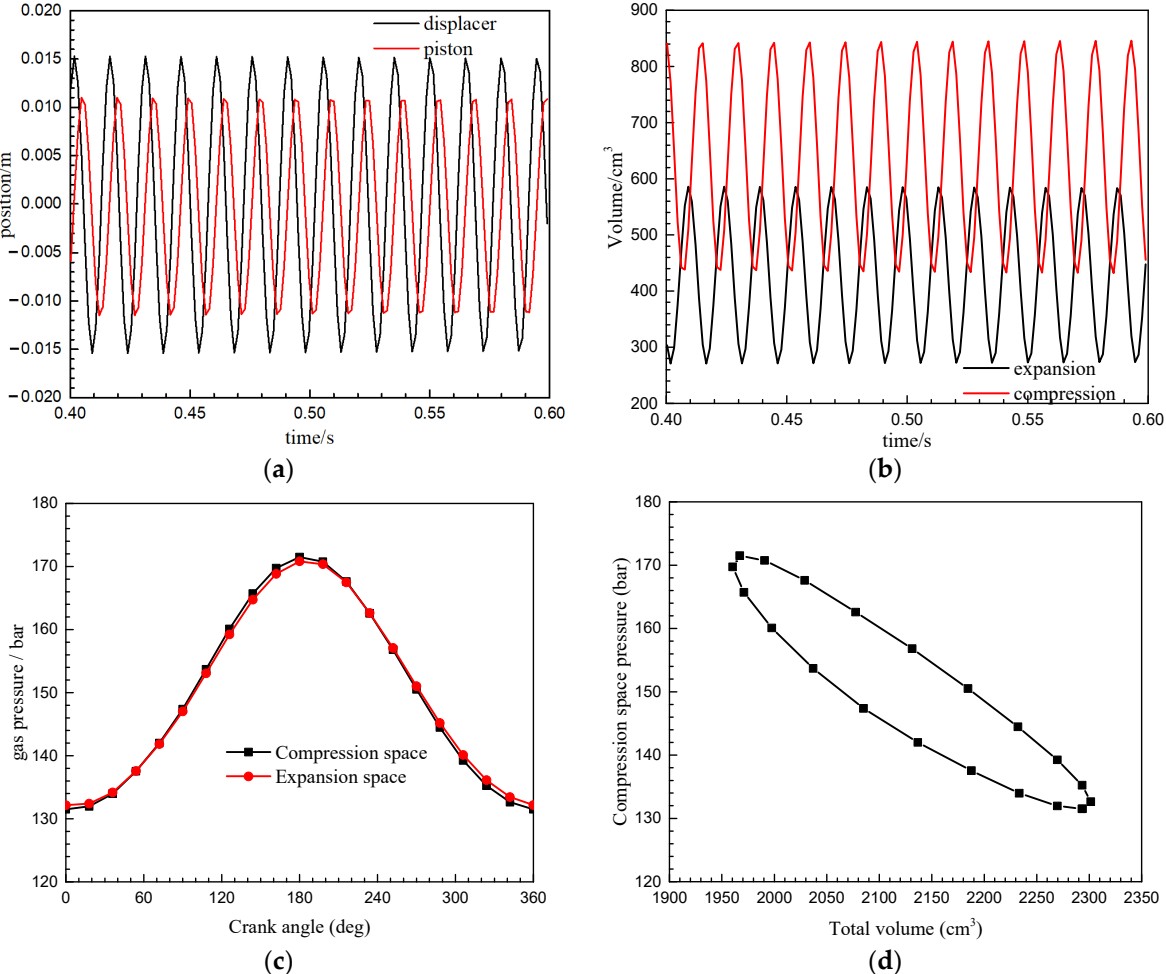

**Figure 8.** Steady-state plots of the CTPC performance: (**a**) position of piston and displacer, (**b**) expansion and compression volume, (**c**) gas pressure, (**d**) PV Cycle diagram under steady-state.

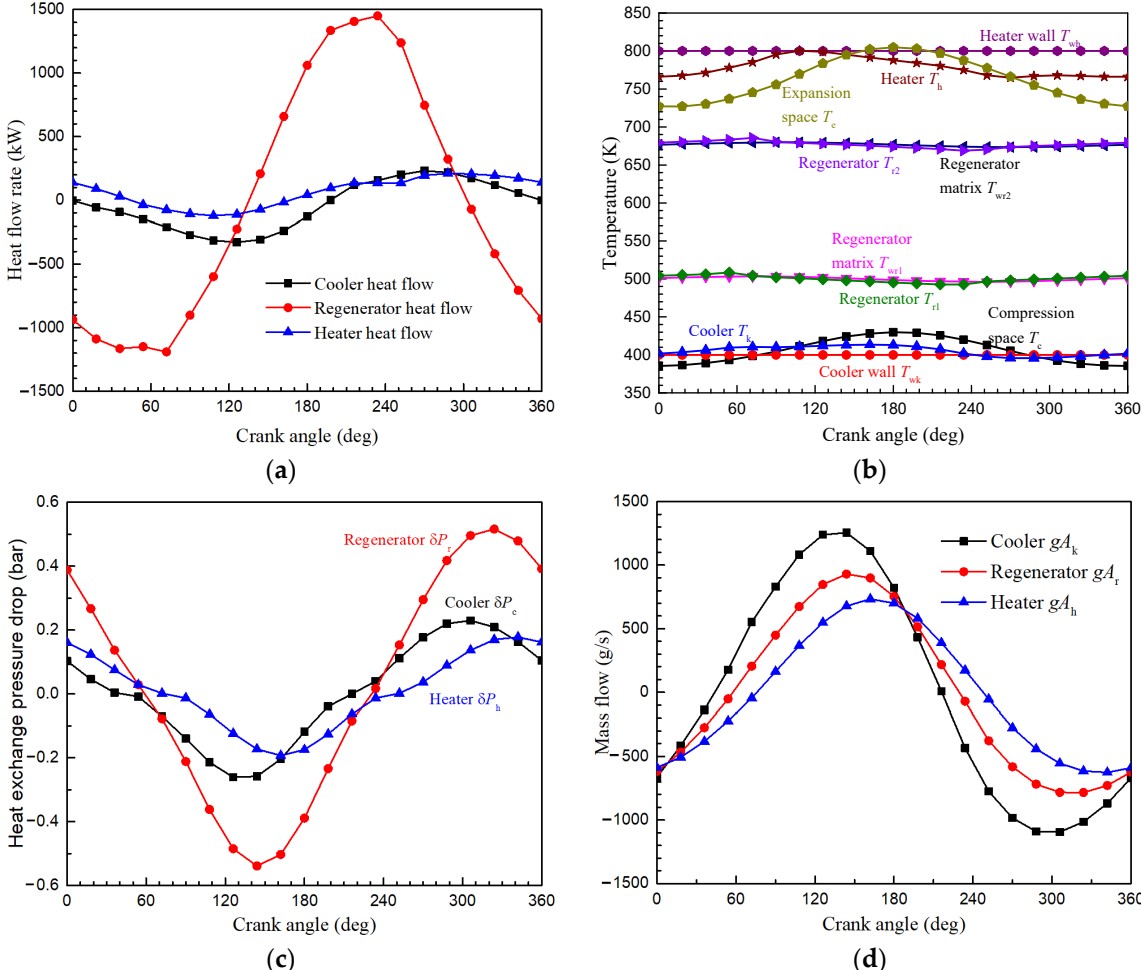

**Figure 9.** Results of the coupled model with losses for CTPC: (**a**) heat-flow rate, (**b**) temperature/theta, (**c**) Pressure drop, (**d**) Mass flow.

The different kinds of heat loss in CTPC versus crank are shown in Figure 10. As shown in Figure 10a, the average heat loss caused by the regenerator is 12 kW, accounting for 64% of the total heat loss. Therefore, the heat loss of the regenerator is very significant, which mainly depends on the efficiency of the regenerator. The heat loss due to irreversibility in the compression and expansion cells is low, which is about 1.26 kW representing 6.8% of the total heat loss. The average heat loss caused by the displacer shuttle effect is about 2.56 kW, accounting for 13.7% of the total energy loss. In Figure 10b, the internal conduction heat losses of heat exchangers are negligible, which are all lower than 0.05 kW. The dissipative heat loss mainly occurs in the regenerator, with an average of 1.57 kW and a maximum of 3.38 kW, accounting for 8.4% of the total heat loss. In the heater and cooler, it's 652.8 W and 568 W, respectively.

The transient current and voltage, as well as the electric power of the CTPC alternator, are shown in Figure 11. The corresponding average electric power output of the CTPC is 17.316 kW and the average heat absorbed by the heater is 66.1 kW, which leads to a converter efficiency of 26.2%. The average current and voltage of the CTPC alternator are 86.38 A and 193.15 V, respectively.

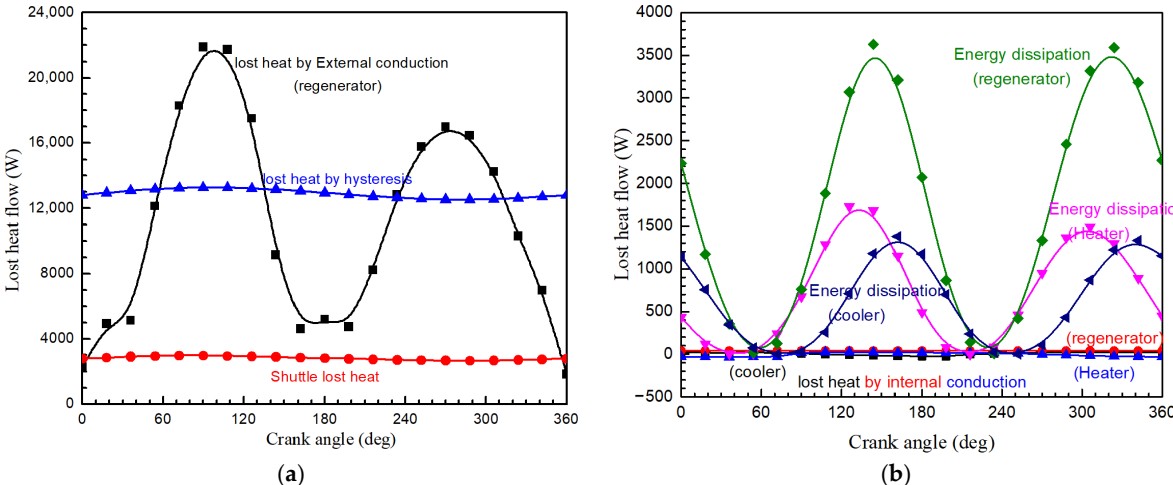

**Figure 10.** Heat loss of CTPC: (**a**) Lost heat flow by hysteresis, shuttle, and regenerator, (**b**) internal conduction heat lost and energy dissipation in the CTPC.

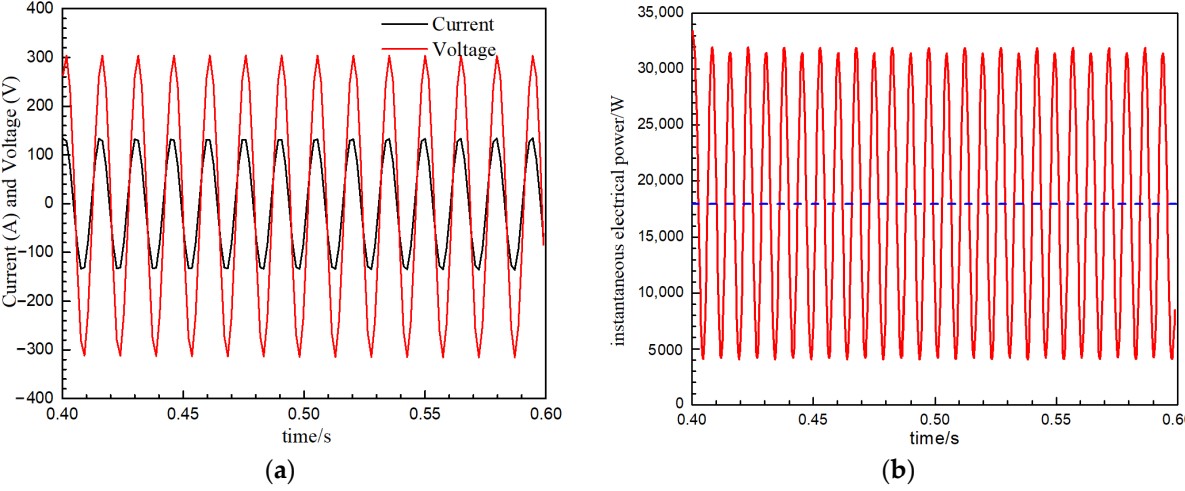

**Figure 11.** The electrical power, current, and voltage for the CTPC alternator: (**a**) Current and Voltage, (**b**) electric power.

### 4.2. Effect of Heater and Cooler Temperature

The performance of SNRPS is mainly determined by the conversion efficiency of its thermoelectric conversion system FPSPC. As FPSPC is a device that converts heat energy into electricity, its conversion efficiency is greatly affected by the temperature of the heater and cooler. Therefore, the developed FPCS_NINT program was used to analyze the effect of the heat exchanges temperature on the system performance of CTPC, including the piston position stroke, energy loss, average electrical power output, and conversion efficiency.

As shown in Figure 12, the position amplitude of the piston and displacer and the temperature of the heater and cooler show a nearly linear correlation. The position amplitude of the piston and displacer decreases with the increase of cooler temperature while increasing with the increase of the heater temperature. For specific CPTC design, the length and space size of the displacer is fixed and the appropriate operating temperature range can be determined. For CTPC design in this work, the length of the piston connecting rod is designed to be 14 mm. Therefore, the optimal temperature of the cold and hot end is 400 K and 800 K, respectively.

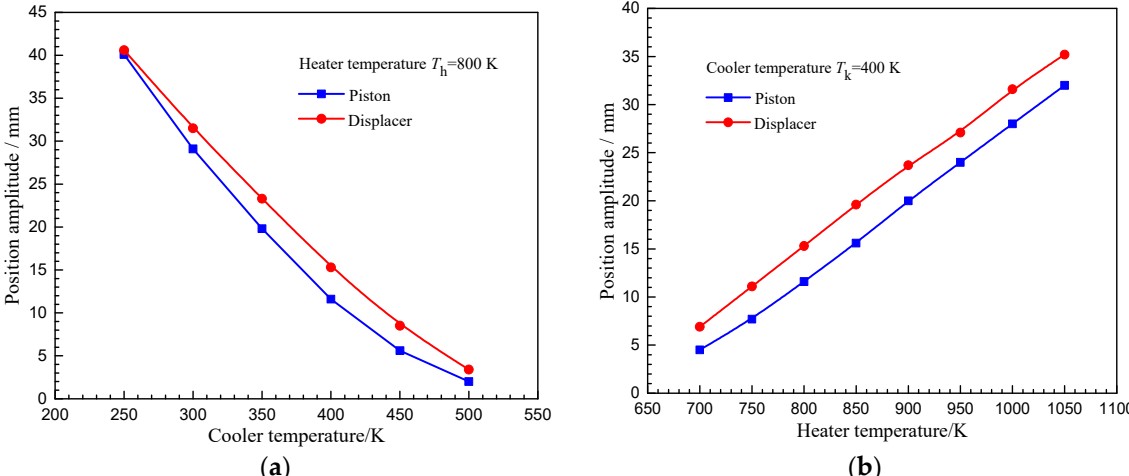

**Figure 12.** Effect on position amplitude of piston and displacer: (**a**) effect of cooler temperature, (**b**) effect of heater temperature.

The energy dissipation in heat exchanges decreases with the increase of cooler temperature while increasing with the increase of heater temperature, as shown in Figure 13. There is an irreversible pressure drop in each heat exchanger, which will cause energy dissipation. As the pressure drop is affected by temperature, the average energy dissipation of the heat exchangers caused by pressure drop is greatly influenced by the hot part and cold part temperature, increasing with the hot and cold end temperature. Besides, energy dissipation is mainly from the loss of the regenerator, because its porous structure results in the large pressure drop loss of gas flow.

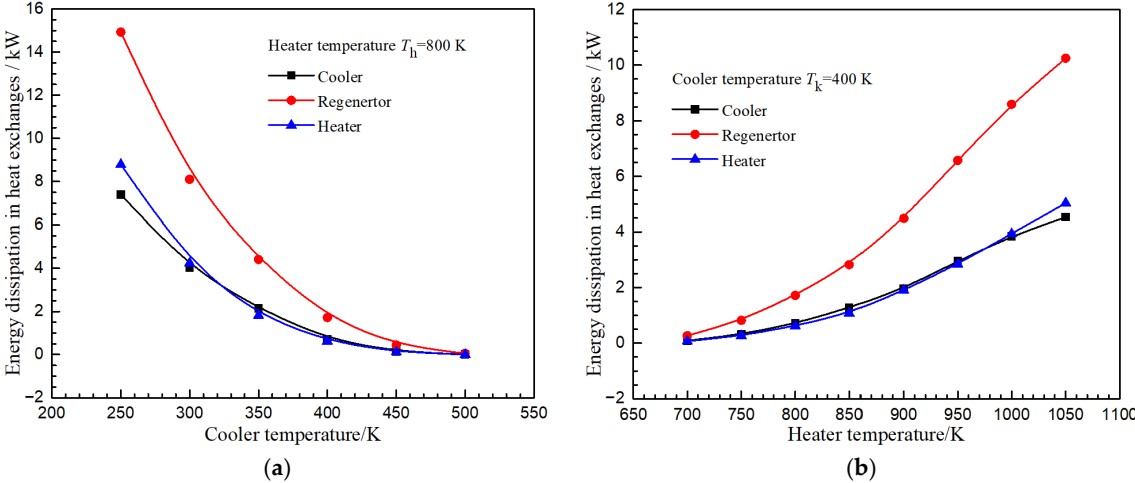

**Figure 13.** The effect on energy dissipation: (**a**) effect of cooler temperature, (**b**) effect of heater temperature.

It can be seen in Figure 14, heat loss caused by internal conduction of heat exchanger is less than 0.1 kW, and therefore it could almost be neglected for the performance analysis of CTPC.

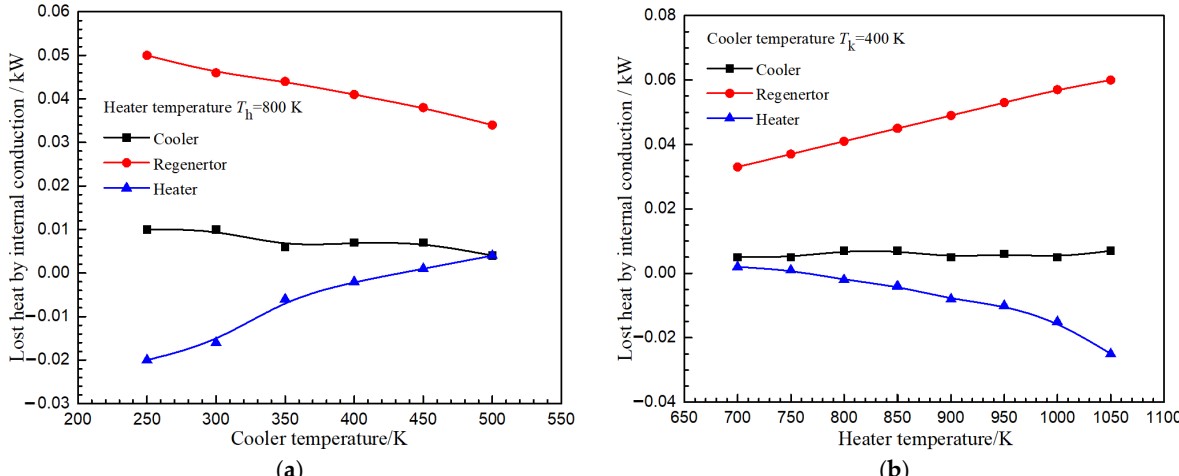

**Figure 14.** The effect on heat loss by internal conduction: (**a**) effect of cooler temperature, (**b**) effect of heater temperature.

The effect on the shuttle heat loss of the displacer and hysteresis heat loss due to irreversibility in the expansion and compression cells are shown in Figure 15. The two kinds of heat loss are both greatly affected by the temperature of the heater and cooler. In general, as the heater and cooler temperature increase, the gas spring hysteresis energy loss increases. The shuttles energy loss of the displacer increases with the increase of the temperature difference between the heater and the cooler. The two kinds of heat losses are larger than 1 kW, which must be considered in the performance analysis of CTPC.

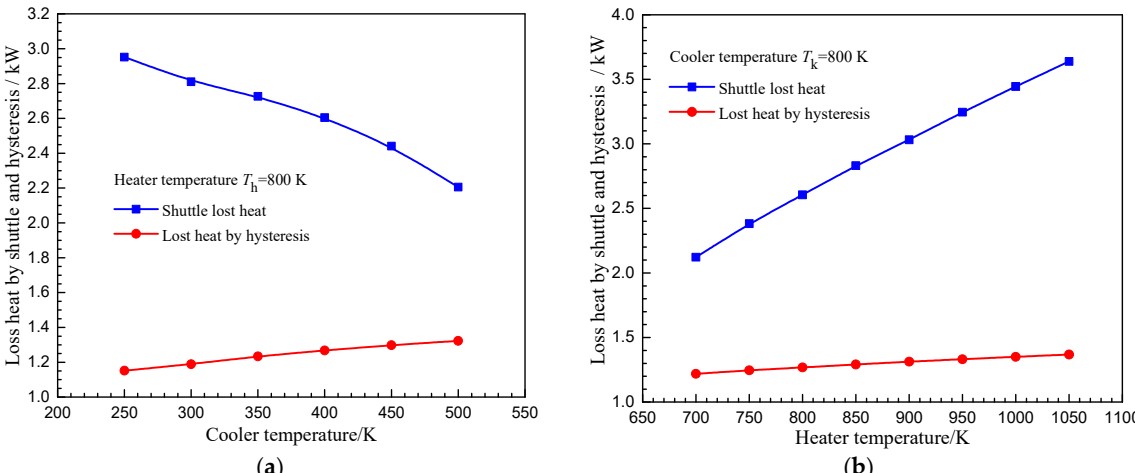

**Figure 15.** The effect on heat loss due to shuttle and hysteresis: (**a**) effect of cooler temperature, (**b**) effect of heater temperature.

The effect of the cooler and heater temperature on average electric power and regenerator heat loss is given in Figure 16. As the temperature difference between the heater and cooler increases, the average electric power output of CTPC increases. But when the heater and cooler temperature difference is less than 200 K, the power output is very low, and CTPC cannot work normally, because the motion of the piston stroke is too small to meet the conditions of reciprocating motion. When the temperature difference between the heater and cooler is greater than 350 K, the stroke of the piston can meet the requirements of power output. As also shown in Figure 16, the heat loss due to heat transfer in the regenerator increases with the temperature difference between heater and cooler, which accounts for the largest percentage of energy loss.

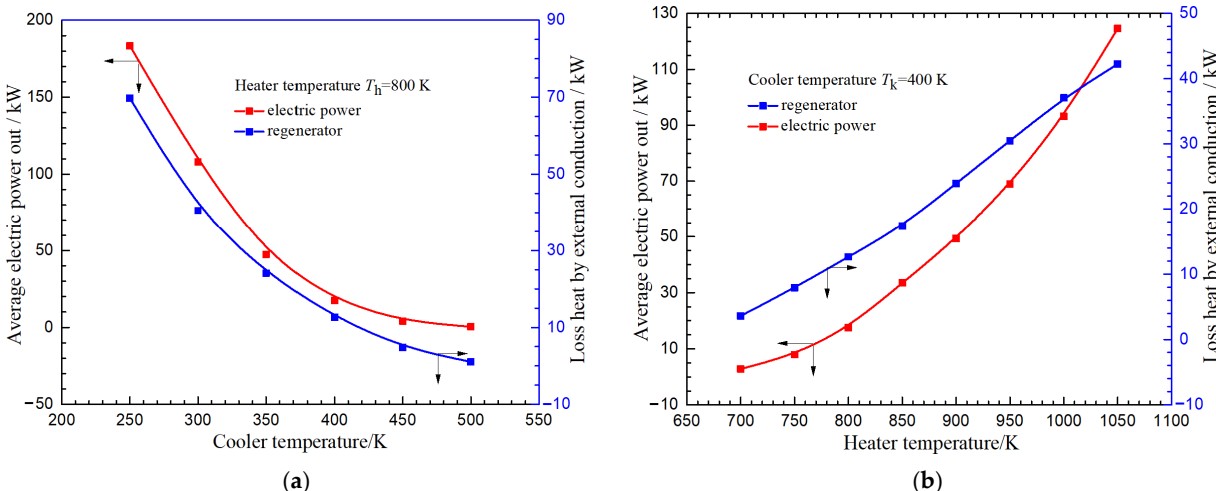

**Figure 16.** The effect on power and energy loss: (**a**) effect of cooler temperature, (**b**) effect of heater temperature.

## 5. Conclusions

A quasi-steady-state thermodynamic cycle analysis model for FPSE is improved by considering parasitic heat losses. By coupling with the piston mechanical movement model and the linear alternator volt-ampere characteristic model, an integrated model is established and a code FPSC_NINT for FPSPC performance analysis was also developed. Both the performance analysis model and the program FPSC_NINT for FPSPC were validated by comparing the calculated results with the experimental data of the GPU-3 Stirling engine. The steady-state performance of CTPC FPSPC developed by NASA was analyzed by using the program FPSC_NINT. The results show that the amplitudes position of CTPC displacer and piston are 15.1 mm and 11.2 mm, respectively. The corresponding average electric power output of the CTPC is 17.316 kW and the average heat absorbed by the CTPT heater is 66.1 kW, which leads to a converter efficiency of 26.2%. The average current and voltage of the CTPC' alternator are 86.38 A and 193.15 V, respectively, which are a little different from those of CTPC design parameter and test data. This is probably due to using different parameters in the model of the alternator and the lack of operating conditions about the test. It is also found that among all kinds of parasitic heat losses, the regenerator heat loss accounts for the largest proportion, with an average of about 12.7 kW. The effects of cooler and heater wall temperature on the performance of CTPC FPSPC were also studied. In conclusion, the program FPSC_NINT which was developed based on a coupling performance analysis model could effectively predict the performance parameters of FPSPC.

**Author Contributions:** Conceptualization, L.G. and H.L.; methodology, H.L.; validation, H.L., S.C. and L.Z.; formal analysis, L.G.; investigation, X.K.; data curation, X.K.; writing—original draft preparation, H.L.; writing—review and editing, X.T.; supervision, J.S. and X.J.; project administration, L.C.; funding acquisition, X.J. All authors have read and agreed to the published version of the manuscript.

**Funding:** This research was supported by the National Key R&D Program of China [Grant No. 2018YFB190064].

**Institutional Review Board Statement:** Not applicable.

**Informed Consent Statement:** Not applicable.

**Data Availability Statement:** Not applicable.

**Acknowledgments:** The authors would like to thank deeply to the anonymous reviewers for their constructive comments on this paper.

**Conflicts of Interest:** The authors declare no conflict of interest.

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
