# Peer review of "Development of a Performance Analysis Model for Free-Piston Stirling Power Convertor in Space Nuclear Reactor Power Systems"

_energies, doi:10.3390/en15030915_

Round 1
Reviewer 1 Report
This article presents a computational approach, based on existing models, to calculate the overall performances of free piston steering power convertors developed for space nuclear reactor power systems.
The authors claim that comprehensive models that includes heat and energy dissipation from multiple sources (in this case the fee piston steering engine, its mechanics, and the alternator) are lacking in literature, providing here an effort in that direction.
The study and its results are both well presented, and conclusions are drawn from the model.
The language and style are appropriate, with only some sporadic punctuation problems and excessively long periods. Nevertheless, there are some critical points that need to be addressed.
Firstly, the different models for each component (piston, alternator) reported in section 2 are taken entirely from cited articles. In some case, whole sets of equations, and even entire figures are directly copied from references, without adequately explicit referencing nor significant changes accounting for the specificities of the studies systems.
It is not entirely clear whether such models can be applied directly to space reactors from the article, and why the choice of such models was made.
I strongly advise to limit the repetitions of cited material to results, and/or cite appropriately (e.g. figures taken from cited articles), as well as describe how such equations apply to the studied case.
Secondly, there is an apparent mismatch between the results of the model compared to direct measurements, and the conclusion of the authors.
For instance, the model discrepancy with the GPU-3 engine test data in terms of output power is about 38%, whereas power and current output calculated from the authors compared to the piston NASA test data are 35 and 79% respectively.
From these data, and from the lack of comparison with other similar or different approaches, it seems hard to claim that the model presented here is an improved integrated model, as claimed by the authors in the conclusions.
I strongly advise to comment and investigate on such discrepancies, and provide other comprehensive models to compare the results with.
In addition to these two major points, there are a few more comments that I would like to report:
- The acronym CTPT is used in the abstract and explained only in Sec. 4.
- The sentence in 1. “Therefore, with the deep expansion…” should have a reference
- In general, Sec. 1. Contains quite strong statements with little referencing backup
- The sentence in 1. Claiming there are relatively few models should be verifiable (see points above)
- There are in general few words on the NINT itself, except a brief description in 3.1
- It is not clear how the crank angle (used as parameter in plots) is linked to the system geometry
- I cannot understand how the variables in Table 3 are connected amongst themselves and with the text/model
- What is the blue line in Fig. 11(b)?
- In Sec. 5, how is this model improved and with respect to what?
Reviewer 2 Report
This paper aimed to model the performance of free piston stirling power convertor, a quasi-steady flow thermodynamic cycle analysis model considering parasitic heat losses has been developed. The performance analysis model has been established by coupling thermodynamic cycle analysis model with mechanical motion model of piston and volt-ampere characteristic model of linear alternator. The analysis model was compared and validated by the experimental data. Overall, this paper is well written, some revisions are suggested to be performed:
1) It is suggested that the author to have a checking for the definition of variables and make them consistent, for instance in Table 1, it is strange that and G both stand for the working gas mass flow;
2) The resolution of some figures needs to be raised, for instance Fig. 4;
3)In section 3.2 “Validation of the model by GPU-3 engine”, the reviewer can’t see what are compared between the model and experiment, please give more details about the validation;
4) There are some typo, the author is suggesed to have a proofreading.
Reviewer 3 Report
The paper is interesting and well-done. The results are motivating. Some points require to be revised:
1-I think the introductory text must be improved. Motivations must be better introduced.
2-The thermodynamical tools used is well-known in literature. Hence, Section 2.1 require some improvement in the style of equations introduced.
3-Figures 4 and 6 are not clear enough.
4-The numerical analysis is well-done but lack more physical discussion in particular to what is connected to experimental studies and results.
5-More perspectives are required.
